# Examination of the ZXY Arena Tracking System for Association Football Pitches

**DOI:** 10.3390/s23063179

**Published:** 2023-03-16

**Authors:** Jon Ingulf Medbø, Einar Ylvisåker

**Affiliations:** Faculty of Teacher Education, Culture and Sports, Western Norway University of Applied Sciences, NO-6856 Sogndal, Norway; einar.ylvisaker@hvl.no

**Keywords:** soccer, football, tracking, measurement quality, validation, telemetry, ZXY system

## Abstract

Modern analyses of football games require precise recordings of positions and movements. The ZXY arena tracking system reports the position of players wearing a dedicated chip (transponder) at high time resolution. The main issue addressed here is the quality of the system’s output data. Filtering the data to reduce noise may affect the outcome adversely. Therefore, we have examined the precision of the data given, possible influence by sources of noise, the effect of the filtering, and the accuracy of the built-in calculations. The system’s reported positions of the transponders at rest and during different types of movements, including accelerations, were recorded and compared with the true positions, speeds, and accelerations. The reported position has a random error of ≈0.2 m, defining the system’s upper spatial resolution. The error in signals interrupted by a human body was of that magnitude or less. There was no significant influence of nearby transponders. Filtering the data delayed the time resolution. Consequently, accelerations were dampened and delayed, causing an error of 1 m for sudden changes in position. Moreover, fluctuations of the foot speed of a running person were not accurately reproduced, but rather, averaged over time periods >1 s. Results calculated from measured values appeared accurate and were readily reproduced in a spreadsheet output. In conclusion, the ZXY system reports the position with little random error. Its main limitation is caused by averaging of the signals.

## 1. Introduction

### 1.1. Background

In association football (soccer or football), two teams of eleven players each move constantly around a pitch that usually is 100–110 m long and 64–75 m wide when used for grown-ups [1]. Their movements are typically characterized by low-intensity motions such as walking, jogging, and slow running, interspersed by periods of high-intensity running [2].

Approaches of analyzing the games may involve analyses of movement of the players [2,3,4,5,6,7,8], tactics, and organization of teams (e.g., [9,10]) or the referees [11,12,13,14]. The technical revolution of electronics and informatics over the last decades now allows recording each person’s position at high time resolution. Several different technical approaches have been tried (e.g., video, global position systems (GPS) or local position systems). It appears that different systems report quite different values, even for the total distance moved, for a player during a match, since differences of more than 10% between apparently reliable systems have been found [7]. When dividing the movements between different categories (walking, jogging, running at different speeds, or sprinting) considerably larger differences have been reported [15,16,17,18,19,20,21,22]. This is important since the size of these errors are comparable to reported differences between leagues and performance levels, between referees, or to reported improvements by time [5,7,8,13,14]. Consequently, it is not known whether reported differences reflect true differences between leagues, level, or by time, or whether the effect at least partly reflects differences between systems and approaches. Computerized systems now allow not only sampling data but also advanced analyses of the data [23,24,25,26,27]. The quality of these analyses is limited by the quality of the data sampled. The quality of commercial tracking systems used to analyze, for example, football games therefore needs further independent examinations. There is, in particular, limited information about to what extent sudden changes in speed or direction are recorded by these systems.

The ZXY tracking system of ChyronHego (Trondheim, Norway; http://chyronhego.com/sports-data/zxy, accessed on 19 May 2017) is a relatively new system for tracking football players. The arena version of that system (ZXY arena) was installed in 2015 at Fosshaugane campus in Sogndal municipality, Norway (61.2316° N, 7.086° E, field altitude 44.4 m above sea level). This system’s performance is addressed here. The basic measurement principle is that transponders worn by the players respond to radio signals being transmitted from beacons around the pitch. The players’ position is calculated from these signals.

### 1.2. Research Issues

Rico-Gonzalez et al. have proposed three sets of criteria for evaluating tracking systems [28]. First, the precision of stationary recordings must be examined. Second, the systems’ (dynamic) responses for speed and accelerations must be examined. Finally, the quality of continuous data registration during play, including a possible influence of nearby players in team sports, also needs to be addressed. We have examined the precision and accuracy of the ZXY system’s measurements on all three aspects proposed by Rico-Gonzalez et al. from the recordings.

More specifically, we have used fixed positions on the pitch and related physical principles as reference and compared the system’s outcome to the expected results. Possible sources of variability or error are addressed. For example, the system transmits radio signals between a beacon, transponders in chips worn by the players, and, for this set-up, four static receivers (radio eyes). Ideally, these signals should go in a straight line through the air only. However, during normal use, human bodies may intercept the paths and thus, perhaps, affect the signal [22,29]. The possible influence of nearby transponders on the reported position of the ZXY system has also been examined.

The system records the position of the chip and, thus, the player at high frequency (20 times per second), but only on filtered signals. This filtering delays the system’s response and may also dampen or eliminate sudden changes (transients), and that effect has also been examined.

The software calculates a number of parameters from the recorded data. It is not known how these derived quantities are calculated. Our former research has shown that some manufacturers of commercial instruments used for sports testing may introduce compensations or modifications, apparently to improve the results [30,31,32]. We have also found built-in errors in calculations [33]. To examine these possibilities, we calculated several of the derived quantities from the measured ones using standard, simple, and well-known equations and principles that can be found in textbooks of numerical mathematics, for example [34].

Results of independent examinations of the performance of the system at Fosshaugane stadium are given here. Our main research question is to what extent the ZXY system records positions and movements on the pitch accurately. We have paid particular attention to how sudden changes are recorded.

## 2. Methods

### 2.1. Description of the ZXY System

The name ZXY refers to the fact that the system may record a player’s position in three dimensions: along the horizontal plane the (XY-plane) and, in addition, the Z-component, recording the position above (or below) the plane. During recordings, each player is carrying a transponder (50 × 36 × 11 mm in dimension, 28 g mass) in a belt worn around the chest or waist. The transponder has a 40-cm long antenna. The first 10 cm of the antenna is bent around the chip, while the distal 30 cm extends roughly linearly away from the chip along the chest or waist belt during normal use.

A beacon sends out signals at 0.05 s intervals to each transponder (signal frequency of 5.3 GHz, corresponding to 5.7 cm wavelength). The reply of the transponder is recorded by four static receivers called radio eyes, each installed on one of the electric pylons at each of the stadium’s corners. There is a delay between the moment the signal is sent to the moment that a reply is recorded by the radio eye, which varies with the distance between the transponder and the radio eye. This allows for the calculation of the transponder’s position. Movement is a consequence of changed position, and when the position is recorded twenty times per second, it may allow recording movements at high time resolution. According to the manufacturer, a direction vector is calculated, and in free space, this vector deviates by, at most, 0.05° (0.001 rad) from the true vector.

The chip also included electronics for gravimetric recordings and accelerations that, according to the manufacturer, are not being used in the present set-up. Likewise, the z-component of the transponder’s position is not calculated. Figure 1 gives an overview of the system set-up for tracking on the pitch.

### 2.2. Experiments

As stated in the introduction, the test should address the precision of the measured position, movements, and possible influence of other players and transponders. The positions of one or more transponders were therefore recorded by the ZXY system at 20 Hz in the following situations:Transponders were placed at known, fixed positions around the field. The system’s reported position and its variation by time was recorded and compared with the true positions.To examine the possible effects of the level above the ground on the recorded positions, two or more transponders were placed at different vertical levels at the same horizontal position, either at the ground, at the seat of a chair ≈45 cm above the ground, or at the seat of a high-rise stool on the chair, being 125 cm above the ground. In further tests, a transponder was mounted on a 3-m long rod that was held up in the air above the field.The possible dual influence of chips was addressed by first placing transponders around the field, well-separated, and subsequently placing one or more transponders close to the first ones.The recorded position of the transponders when a person’s body intercepted the path between the beacon-radio eyes and a transponder was compared with that of the transponders with free sight lines to all radio eyes.

Figure 2 gives an outline of the main idea of the experiments.

A system recording at 20 Hz would expectedly reproduce fast changes in position and speed, but filtering may dampen the response. We carried out several tests to examine the effect of filtering on the data output:During running, the speed of the foot relative to the ground varies during a stride cycle, even if the speed of the body is constant. To examine to what extent the ZXY system reproduces these fast changes in speed, transponders were mounted around the ankle, in addition to one around the waist, of a test person running across the field, and the data were compared.A transponder was mounted close to the nipple at the inner side of the rim of a bicycle wheel, and the cycle was walked at a low, constant speed across the field. The speed of the transponder would, in the time-X-plane, be close to the sum of a constant component and a sinusoidal curve (in the XZ-plane, a curtate trochoid) superimposed upon it. It was examined to what extent filtering the signal changed the response.A person wearing a transponder moved to a defined line on the field before returning, either after a few seconds rest at the line, or immediately with no rest. It was required that both feet were placed at or beyond the line before returning. The system’s recorded minimum distance from the line was registered.

The experiments were carried out mostly during cloudy weather conditions with no precipitation and at air temperatures of 10–20 °C. The principal investigator (first author), being fully aware of his rights, served as test person. The study was approved by the review board of the Faculty of Teacher Education, Culture and Sports (FETIKO) at the university.

### 2.3. Analyses, Calculations, and Statistics

The data array was imported to an Excel spreadsheet and converted so that each parameter was displayed in separate columns. Speed, acceleration, distance moved, angle and direction of movement, and speed of cyclic movements were taken, as further detailed in the mathematical Appendix A. In short, we used standard equations found in most textbooks of numerical mathematics (e.g., [34]).

Variations in recorded parameters by time have been reported as the standard deviation of series of values of the parameter in question.

## 3. Results

### 3.1. Output of Results

The ZXY system reports the results in standard SI units, using a coordinate system where the origin is at one of the field’s corners. This means first that the position is given in meters and with seven-digits precision (five to six decimals). The x-coordinate runs along the field, the y-coordinate gives the position across the field, and the z-coordinate runs vertically. The exact time point at measurement is based on UCT (universal coordinated time) and is reported as Unix time in seconds with five decimals [35]. That time measure is readily converted to a common and easily readable date and time format, usually provided by ZXY’s software.

The software further calculates and reports the distance covered from the start in meters, the speed in m s^−1^, the acceleration in m s^−2^, and the instantaneous direction of movement as an angle in radians (−π, π]. The output also includes two parameters called “heading” and “turn speed”, respectively. Heading reports the orientation of the chip. If properly mounted on a player moving forward, the value of heading equals that of direction. If the player moves sideways or backwards, the two parameters will differ by π/2 (90°) and by π (180°), respectively. Turn speed is the time derivative of the heading (change in heading by time).

A standard output file also includes the heart rate and a parameter called effort. These two quantities are not addressed further here.

### 3.2. Quality of Recorded Values

#### 3.2.1. Time

During the first series of measurements, the system’s clock was around 35 s behind standard time as taken from that given by the Norwegian Metrology Service (no. Justervesenet). This problem was addressed to the manufacturer who reset the system clock, and during the subsequent sessions, no detectable difference was seen.

#### 3.2.2. Drift at Onset and End

As the system is switched on and the recordings start, there is usually a drift of ≈0.1 m in the reported position. Within 1–3 s, the drift ends (Figure 3). A corresponding effect is seen when the system is switched off. Since these transient effects seem to be of, at most, a few seconds duration, the only precaution needed when analyzing the data is to discard the first and last few seconds.

#### 3.2.3. Precision of Stationary Recordings in Free Air

Transponders were laid on the field, thus being motionless, for at least 1 min, and the positions were recorded. The value of the recorded position varied, usually by 2–3 cm (standard deviation), but sometimes considerably more (10–20 cm). It appeared that averaging the values over 1 s did not reduce the variation much, which would have been the case if the variability was due to random, independent events. A further examination of the recordings showed regular fluctuations (Figure 4). The variability was largely caused by fluctuations with cycles of several seconds duration (≈0.3 Hz). This explains why averaging over 1 s had minimal effect on the variability.

##### Variability between Transponders

Transponders were placed at known positions at the field, and the recorded coordinates were then compared with the known coordinates. The values differed, sometimes systematically (see Figure 5), and sometimes with no apparent relationship to the known position (not shown).

The variability in the bias (error of regression) was 0.1–0.2 m in one dimension. A similar variability was also found for the other coordinate. The pooled errors of the two coordinates correspond to an error in the xy-plane of ≈0.2 m. 

##### Recordings of Vertical Position (z-Coordinate)

The outputs did not report the vertical position accurately. For example, for transponders laid on the ground, on the seat of a chair ≈45 cm above the ground, and on the seat of a high-rise stool placed on the chair, ≈1.25 m above the ground, the reported z-coordinate varied between 0.92 and 1.00 m above the ground, irrespective of the actual position (ground, chair, or stool on chair). In addition, the reported xy-coordinates differed by up to 0.3 m when the z-position was changed, despite no displacement in the xy-plane.

For a further examination, a transponder mounted at one end of a 3-m-long rod that was raised up in the air to a level >5 m above the field, held there for some seconds and repeatedly raised up and down in slow movements. Those changes in vertical position were not reflected in the system’s output.

#### 3.2.4. Time Resolution of Changed Position—Possible Detection of Transients

To examine the effect of filtering, experiments with sudden changes in position and speed were carried out to see whether the system reproduced these changes accurately.

A test person ran across the field while wearing transponders around the ankle of each leg. As the foot was on the ground in the stance phase, it barely moved (relative to the field; ≤15% of the forward speed of ≈5 m s^−1^, corresponding to ≤0.75 m s^−1^), while in the swing phase, it moved much faster than the runner. These fluctuations were not well recorded by the system (Figure 6). For example, for a subject running at 5 m s^−1^, the reported calf speed varied by 0.13 m s^−1^ (standard deviation, coefficient of variation 2.6%). In particular, the recorded speed was never close to zero, as would have been expected during the stance phase for nonfiltered values. Likewise, for a person walking at 1.5 m s^−1^ with a stride length of 0.9 m, giving a stride cycle of 0.6 s duration, the reported calf speed varied by 0.16 m s^−1^. The picture was similar to that in Figure 6, but with a mean speed ≈1.5 m s^−1^ rather than ≈5 m s^−1^ (not shown).

In further experiments, a transponder mounted on a bicycle wheel was walked across the field at 1.5 m s^−1^, thus making one full revolution in ≈1.5 s. The expected forward speed of the transponder would be sinusoidal, with minimum values of 0.3 m s^−1^ and peak values ≈2.7 m s^−1^. The observed fluctuations seen were only 19% of the expected ones (Figure 6, lower panel). This shows that the built-in averaging in the system’s output does not allow detecting changes in the time order second, despite the 20-Hz output.

The speed curves in Figure 6 (lower panel) were integrated over the time, giving the distance covered. The two integrals did not differ systematically. Admittedly, over short distances (or time periods) covering only a fraction of a full wheel cycle, one integral might be up to ≈0.5 m ahead of the other. Before the full cycle was completed, the other integral caught up with the leading one, within an error of measurement of ≈0.1 m. This means that the signal filtering did not introduce systematic errors in the recorded distance beyond the effect of filtering itself.

To examine the effect of filtering further, a test person wearing a transponder moved to a marked line on the field and returned. The recorded position of the turn point varied depending on the pattern of movement (Table 1).

The reported values are the closest distance from the line of turn as reported by the ZXY system. In all experiments, the test person placed both feet at or just beyond the line before turning.

It appeared that if the test subject just passed the line with his feet before turning immediately, the system’s closest recorded position was ≈1 m from the line. The mismatch between the true and the recorded position did not differ much between walking and fast running as long as the turn was carried out immediately after both feet had passed the line.

#### 3.2.5. Site of Measurement

In normal use, the tip of the transponder’s antenna extends 30 cm away from the chip. Since the spatial resolution under conditions in the measurements above sometimes was considerably better than this, it was attempted to examine what part of the transponder (chip or antenna) decided the transponder’s recorded position. Transponders were therefore laid on the ground with the chips at the rim of the penalty field and antennas pointing towards the middle of the field, and the position was recorded. The transponder was subsequently rotated half a revolution around a vertical axis so the antenna was pointing towards the goal line, leaving the chip in the same xy-position. The transponders’ reported positions were recorded again. It appeared that it changed by around 0.3–0.4 m, but with some variation (Figure 7). This suggests that the recorded position cannot be ascribed to one specific part of the transponder (e.g., chip or tip of antenna) but rather being an average of the transponder’s components’ position.

In these experiments, the y-coordinate of the tip of the antenna of each transponder was not changed when the transponder was rotated a half revolution around the z-axis. Nevertheless, the reported y-coordinate changed as much as the x-coordinate changed (Figure 7, lower panel).

#### 3.2.6. Possible Influence of Human Bodies

For ideal recordings, there is free, direct sight line between the beacon-radio eyes and the transponders. In real use, the chips are worn on a human body that may screen the chip from at least one radio eye. Moreover, other players may be close by. Thus, the possible effect of the presence of a human body that intercepts a straight-line view through the air between the transponders and radio eyes was examined.

A transponder was laid on the ground, and at time zero, the test leaders laid down above the transponder, screening it without physical contact. The reported position became more variable, and sometimes it changed by more than 0.2 m, even when averaged over a 10-s period (Figure 8).

#### 3.2.7. Possible Interference of Nearby Transponders

A transponder was placed on the ground, and the position was recorded. Another transponder was laid close to the first one, and the position of the first one was recorded again. Sometimes the signal became more noisy, and occasionally, a systematic change in position was seen. The main picture was nevertheless that if a systematic change was seen, the effect was less than 0.2 m.

### 3.3. Quality of Calculated Results

The ZXY system reports a number of calculated quantities provided as an Excel file. The accuracy of these calculated values has been examined. 

#### 3.3.1. Speed

The speed calculated from the change in position data was compared with the system’s reported values. Only differences of around 0.02 m s^−1^ or less were seen. Moreover, sometimes the system’s values were slightly larger than the calculated ones, and at other times, the difference was reversed. There was thus no systematic difference over time. Admittedly, over short time periods of less than 3 s, there could be small, systematic differences, which means that the small differences seen were not fully random.

#### 3.3.2. Acceleration

The acceleration was calculated and compared with the system’s reported values. In a few, single cases differences up to ≈0.2 m s^−2^ were seen. In most cases the differences were considerably less. There were no systematic differences in the two series of acceleration data when examined over longer time period. As for the speed data, there could be small systematic differences over short time periods (a few seconds).

#### 3.3.3. Distance

Distance covered is the speed integrated over time. The distance covered was accordingly taken from the calculated speed and compared with the system’s corresponding value. Only minimal differences were seen. For example, for a 100 m run, the two results differed by less than 0.1 m.

The results presented above show that the reported position of motionless transponders fluctuates. Moreover, a rotation without translation may result in changed xy-coordinates. These effects are taken as movement by the system. Six transponders lying motionless for more than 15 min apparently moved 2.7 ± 1.1 m/min (4.5 ± 1.8 cm s^−1^). This translates to 244 ± 97 m during a 90-min football match.

#### 3.3.4. Direction

The direction of movement was calculated and compared with the system’s reported values. During movements, the values differed by no more than 0.002 rad (0.1°). At periods when the subject hardly moved, differences up to around 0.025 rad (≈1.5°) were occasionally seen.

### 3.4. Summary of Experiments and Results

Our main results are summarized in Table 2.

## 4. Discussion

The main results in this study are first that spatial resolution of the ZXY system is ≈0.2 m. Second, although the system records data at 20 Hz and reports data at that time resolution, fast transients are not recorded accurately. Third, there were no significant errors in results calculated from measured values.

### 4.1. Principles of Evaluation

The ZXY system is one among many to record positions and movements in team sports. We have used physical positions on the pitch and further known movements to which the outcome of the ZXY system has been compared. Our examination has addressed three sets of criteria for evaluating tracking systems that have recently been proposed by Rico-Gonzalez et al. [28]: precision of stationary recording, dynamic responses, and possible influence of nearby players. This approach is at variance with recent propositions from FIFA suggesting that tests should be restricted to (simulated) matches [36]. However, if mismatches are found, it is important to know the cause of the mismatch (static, dynamic, or influence by other players), which has also recently been emphasized by Aughey et al. [37]. In addition, FIFA’s proposed statistical criteria may need a re-evaluation. We have, in addition, addressed the quality of calculated results.

### 4.2. Precision of Stationary Measurements

For motionless transponders, the recorded position fluctuated somewhat, typically with a frequency of around 0.3 Hz. The reason for these fluctuations is not known. A former study has also observed positional fluctuations from transponders on persons standing still [17], perhaps caused by small sways of the body position during standing. Similar fluctuations were seen in our study from transponders lying on the ground. 

The radio eyes were mounted on high-rise pylons. It is well known that high-rise structures fluctuate in the air as a combined effect of wind and the structure’s mechanical stiffness. It could be that the observed fluctuations reflect such movements and thus have a simple mechanical cause. If so, the effect may be corrected for, and the corrected signal may become considerably more precise than that observed here. It could also be that the fluctuations seen were artifacts, for example, of electronic origin (e.g., aliasing). We have limited insight into properties of the stadium’s mechanical structure and function as well as to the ZXY system’s electronics and data handling. 

There were also some deviations between the known and the reported positions. In general, the errors, both random variations and systematic errors, were 0.2 m or less. That is in line with the system’s claimed accuracy (personal communication with ZXY staff). Sometimes, the variability was considerably less than 0.2 m. Human bodies intercepting the signal may affect the reading [29], but again, the error was ≈0.2 m in our study. Likewise, there were no major effects of nearby transponders, as also suggested by a former study [22].

There is a small drift in the signal at the start and end of recordings, at most of a few seconds duration. This may be related to the filtering of signals addressed below. Since the system is normally started long before the start of a match or a training session and runs to its end, the drift will expectedly have no significant influence on the results.

The conclusions above are based on analyses of filtered data. The reliability of unfiltered (or less filtered) data cannot be assessed from our data.

### 4.3. Quality of Calculated Results

The ZXY system provides a number of derived quantities from the measured quantities (e.g., speed from changed position, acceleration from changed speed). It is not known how these values are calculated. We therefore calculated several of the derived quantities from the measured quantities using standard, well-known equations and principles that can be found in textbooks of numerical mathematics, for example [34]. In all cases examined, our calculated values were very close to those calculated by the ZXY system’s software. Consequently, as long as ZXY’s raw data are reliable, so are those derived from them as well. Admittedly, we did not examine the quantities called, “heading”, “turnspeed”, or “effort”, nor have we examined the reported heart rate.

The comparison of our calculated quantities with those reported by the system has further consequences. We calculated the speed and acceleration as the first and second derivative of position by time, respectively. The system can also measure the acceleration directly from sensors within the chip, that is, by a measurement independent of the position (not used for the present set-up according to the manufacturer). An alternative approach could be to take the speed as the time integral of the acceleration, and further the distance moved (or position) as the time integral of speed. It is well known that numerical differentiation may be highly unstable and, therefore, non-robust, particularly for time steps as short as 0.1 s or less ([34]: 504). Conversely, numerical integration is known to be a robust process, largely eliminating random effects (noise) in single input values unless a bias is present as well. These mathematical relationships between two independent measurements would allow independent verification and control of the quantities in question.

The conclusion so far is that once data have been sampled and accepted by the system as reflected in the system’s output files, the further processing appears to be with no essential error.

### 4.4. Time Resolution—Dynamic Responses

The ZXY system outputs only filtered data. This means that the reported “raw” value seen in the output files is an average of many readings, probably introduced as a means to reduce possible noise. An undesired side effect is that the time resolution is reduced. In addition, if the filtering (averaging) is done in real time, the reported values lag behind the true values. The consequence of the filtering was first that cyclic movements was not well reproduced by the system but rather largely “averaged”. A further consequence is that when the true movement is not in straight line, the recorded path will cut corners and thus be shorter than the true path [38]. Our measurements show that this effect may be considerable for the ZXY system, up to around one meter, for fast, sudden changes in movements. This again means that the system presumably underestimates the true distance covered during nonlinear movements, but to an unknown extent.

We did not record speed or acceleration by independent measurements. However, the discussion above suggests that the reported values of speed and acceleration calculated from filtered (averaged) data do not reflect the true speed or acceleration accurately unless the values remain constant for several seconds. This is a major limitation of the system.

### 4.5. Comparisons with Other Systems

The ZXY system is one of many that provides data on players’ movements on the field. The first approaches used were camera-based or video-based time–motion analyses of games, e.g., [7,39]. Such systems are now computerized and based on many cameras. Another quite commonly used approach is based on the GPS technology [7,12,15,16,17,28,40,41,42,43]. The performance of such systems has been addressed in several studies, providing conflicting results and conclusions. More recently, local position systems have been introduced [17,18,21,22,28,29,38,44].

Camera-based systems may provide highly reliable data for simple movements like sprints in a straight line [39]. However, that kind of simple movement is not typical for ball games. When the performance of two different systems were compared for the same football games, differences of 10–50% were found, depending on the type of movement examined (jogging, running, or sprinting [7]). More recent studies have found that errors for video-based systems are 2–4 times larger than for local position systems [18,22].

Possibilities and limitations of GPS-based systems have been addressed [40,45]. These systems may apparently provide position data with errors in the centimeter range under ideal conditions. However, when used in sports testing, quite large errors are found. For example, during a sprint, the errors may be more than 10%, even if the sampling frequency is as high at 10 Hz [40]. The errors may be quite large, even for runs of longer distances, and the error increased by the speed [7]. This latter effect is important since it is, in particular, runs at high speeds that are of importance when analyzing football games [3,5]. The sampling rate is also important since the reported distance moved during a football game was apparently 10% less when analyzed from data sampled at 1 Hz compared with 5 Hz sampling rate [7]. For running at high speed, the relative differences were much larger.

One problem is systematic differences between systems. This means that at least some of them do involve quite large systematic errors, in particular, for runs at high speeds. Another problem is the random error or variability. Edgecomb and Norton found that the variability of the two systems they examined (a GPS-system and a computerized tracking system not specified further), was 5–9%, taken as the technical error of measurement, and both the intratester and the intertester variabilities were of similar size [41].

A number of studies have compared the performance of local position systems with that of other systems [17,18,28,44]. These systems seem to perform better than other technologies (GPS and video-based systems).

We provide data on independent verification of calculated results, which appears not to have been included in former studies. Further, there is limited data in metric units, as are used in this study. That makes direct comparisons more difficult. The spatial resolution of the ZXY system is around 0.2 m. Some systems using local position technology perform at that level [18,21,22,37] or somewhat better [19,29]. Many other systems do not perform that well [15,16,17,20,22,28,40,41,42,44]. Thus, the performance of the ZXY system in the present study seems to be among the better ones, as judged from the examinations of instruments discussed above.

### 4.6. Need of Further Work

The discussion above shows that further research on systems’ performance is needed. There is, in particular, a need for more standardized tests. Future work needs independent verification against physical measures as used in this study, against high-quality (‘error-free’) measurements as may perhaps be provided by some recent video-technology systems [18,19], or against high-quality measurements available for simple movements. One possibility may be to record the position of players that run along known paths that mimic those of a football player in a real match. These paths will thus not be simple, straight-line paths, as in another study [39], but rather include sharp turns around corners, as proposed by Frencken et al. [38]. The length of the paths must be accurately known, and the system’s reported distance covered must be compared with the true distance.

Further, players may run at known speeds to allow the system’s reported speed to be compared with the true speed [5,42]. Since fast runs appear to be most decisive in a game, high-speed runs are of particular interest. Accelerations may be tested in linear runs against high-quality measurements (e.g., radar or laser), perhaps also against measurements on force platforms. The effect of filtering of raw data needs further examination. We agree with Linke et al. [18] that low-pass filtering with a cut-off frequency of ≈2 Hz may be a promising approach. A higher cut-off frequency may include accelerations and deceleration within a stride [37], which, in most settings, are not wanted. 

Aughey et al. recently proposed that a precision in position better than 0.2 m is not needed, referring to the size of body dimension [37]. That may be acceptable for (static) measurements of position. However, speed and other quantities calculated from the position may need better precisions in position data. This issue needs further examination.

Some systems, including the ZXY, show fluctuations in positions at rest that appear to increase the total distance covered. This effect needs further examination.

The ZXY system defines movements less than 2 m s^−1^ as walking, speeds up to 4 m s^−1^ as jogging, then up to 5.5 m s^−1^ as running, faster than that, up to 7 m s^−1^ as fast running, and above 7 m s^−1^ as sprinting. This is a classification proposed by Rampinini et al. [46]. Other systems may have the same (e.g., [2,5]) or different definitions (e.g., [4]). Different definitions make comparisons of results from different studies difficult. We suggest a standardization of the cut-offs between different types of movement, preferably based on functional classifications inherent to the game. For example, the cut-off between jogging and running of 4 m s^−1^ means that completing a full marathon run in 3 h (at an average speed of 3.9 m s^−1^) is classified as jogging.

Analyses have so far largely been carried out on grown-up men. Participation in women’s football has been increasing over the last decades and presumably will continue to do so in the future. Consequently, movement analyses may be of interest for women’s football too. They do not run as fast as men, and their different movement categories may need cut-offs different from those appropriate for men’s games, as has been proposed by Park et al. [47].

The systems addressed in the discussion above have mostly been used for analyzing soccer games. It is conceivable that they may be well suited for analyzing movements in other sports too.

## 5. Conclusions and Practical Application

The ZXY tracking system appears to be suitable for recording players’ position around the field during a game, and thus, to provide data on movement and running. A major limitation is that filtering the data during the sampling and processing reduces the time resolution. The system nevertheless appears to perform at least as well as most other tracking systems, in particular, because of its high spatial resolution.

## Figures and Tables

**Figure 1 sensors-23-03179-f001:**
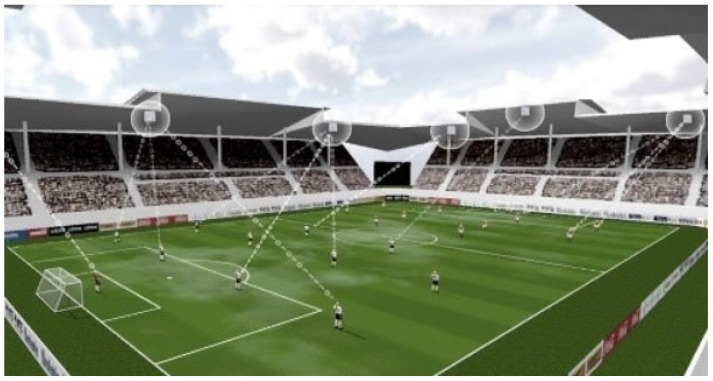
Principles of tracking by the ZXY system. Transponders worn by the players respond to signals sent out from beacons at known positions around the pitch. The players’ positions are calculated from the recorded replies. Photo taken from the ZXY brochure, also available at https://www.nacinc.jp/analysis/sports-tracking/zxy-arena/, accessed on 10 March 2023.

**Figure 2 sensors-23-03179-f002:**
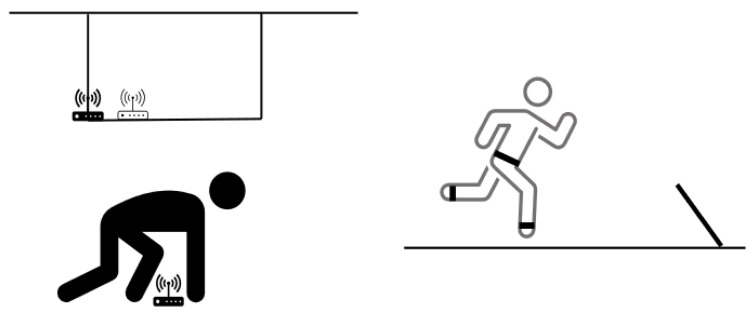
Outline of the experimental set-up. Left, static measurements and possible interactions. Upper part, a transponder (black) was placed at a known position on the field. The system’s reported position was recorded and compared with the true position. A possible influence of a nearby transponder (white) was also recorded. Lower part, possible influence by a human body was recorded separately. Right part, dynamic measurements. A person walked or ran to a line at known position (dashed), and the system’s reported path was compared with the true position. In further experiments, the person wore transponders around the ankle, in addition to one around the waist (black bars). The reported movements were compared with those expected.

**Figure 3 sensors-23-03179-f003:**
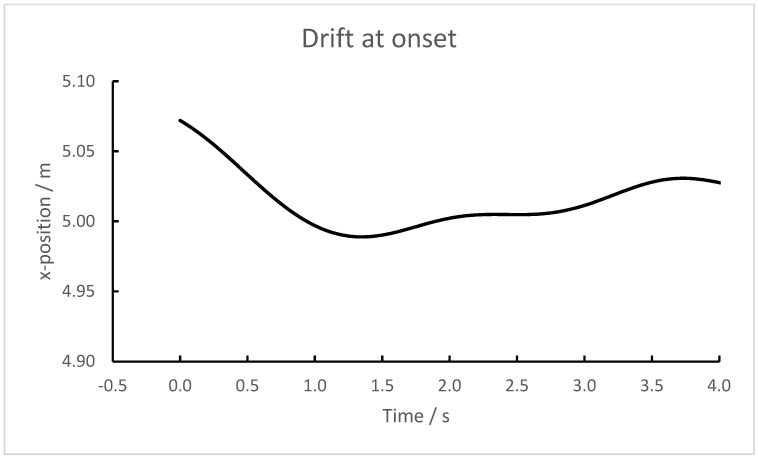
Recordings of the x-coordinate of a transponder lying on the ground during the first 4 s of a recording. Note that the y-axis does not start at zero.

**Figure 4 sensors-23-03179-f004:**
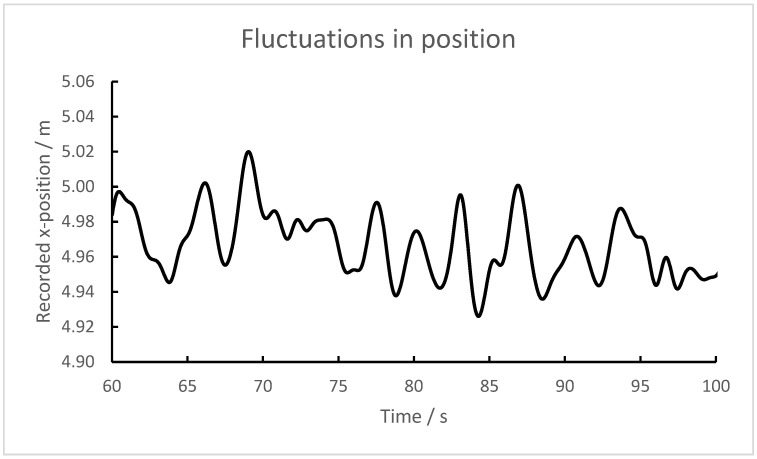
A 40-s recording of the x-coordinate of a transponder lying on the ground. Note that none of the axes starts at zero.

**Figure 5 sensors-23-03179-f005:**
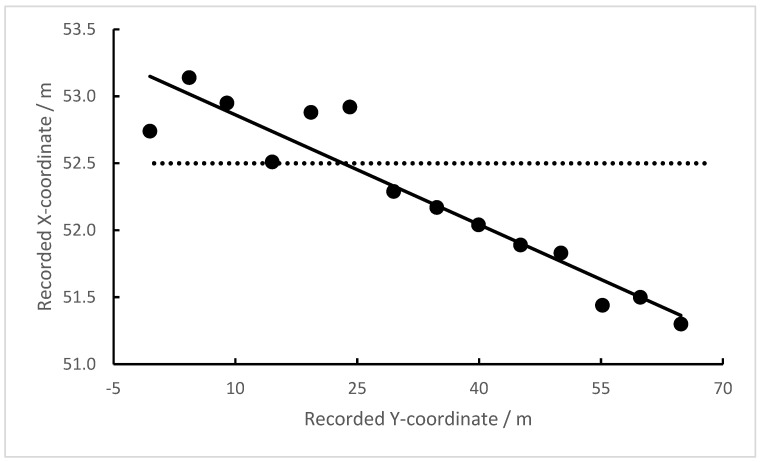
Recordings of the x-coordinate of transponders lying on the ground along the field’s midline. The dashed line is the known x-coordinate.

**Figure 6 sensors-23-03179-f006:**
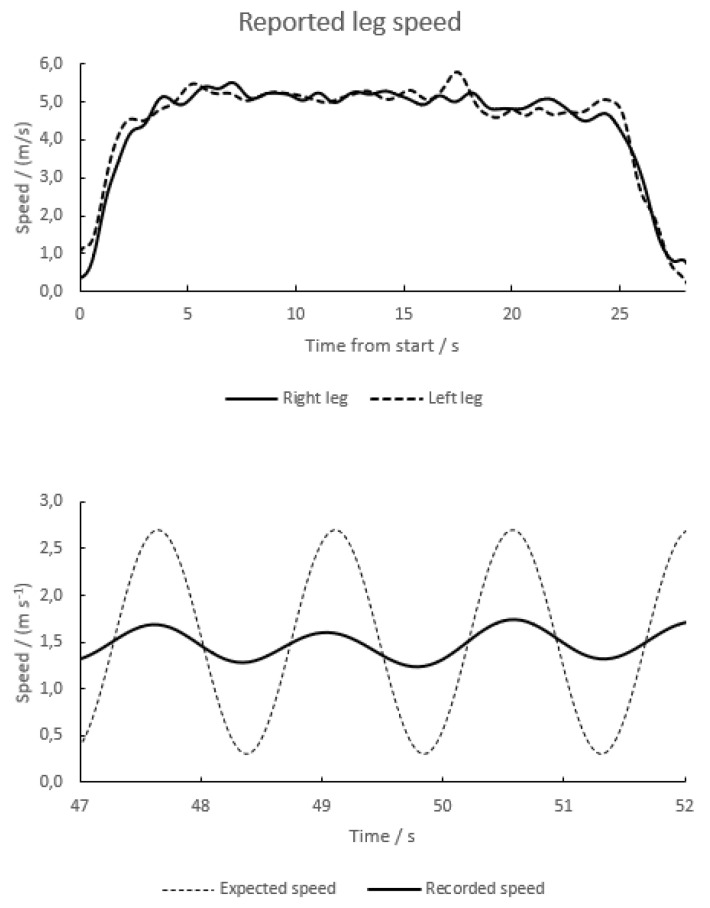
Upper panel, recordings of the speed of transponders being tied around the ankle of a test person running across the field at ≈5 m s^−1^. The two curves are for the right and left legs, respectively. Lower panel, recordings of the speed of a transponder tied to the rim of a bicycle when rolled at constant speed across the field at 1.5 m s^−1^, giving a time of ≈1.5 s/wheel cycle. The dashed curve gives the expected value of the true, unfiltered speed.

**Figure 7 sensors-23-03179-f007:**
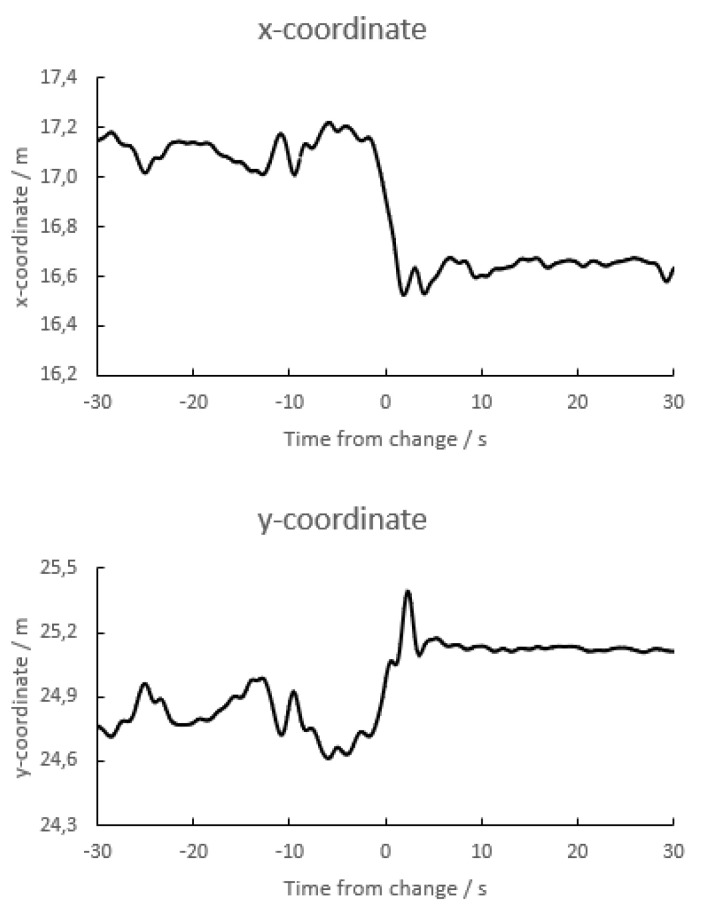
Recordings of the x-coordinate (upper panel) and y-coordinate (lower panel) of a transponder lying on the ground and being rotated half a revolution around the z-axis at time *t* = 0. During the rotation the x-coordinate of the antenna’s tip changed 60 cm while the chip was in the same position. There were no changes in the true y-coordinates. Note that the y-axes do not start at zero.

**Figure 8 sensors-23-03179-f008:**
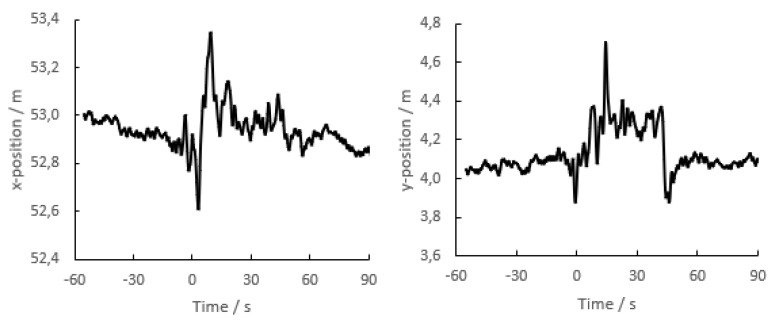
Recordings of the x-coordinate (**left**) and y-coordinate (**right**) of a transponder lying freely on the ground. At time zero, the test leader laid down above the transponder, screening it with his body. At time 45 s, the person stood up, leaving the transponder uncovered. Note that the y-axes do not start at zero.

**Table 1 sensors-23-03179-t001:** Recorded position of turn point for a test person moving to a defined line and then returning.

Test No.	Movement	Recorded Distance from the Line/m
1	Walked to the line, 5 s rest before returning	0
2	Walked to the line, sudden turn and return with no rest	0.78
3	Jogged to the line, sudden turn and return with no rest	0.93
4	Ran to the line, sudden turn and return with no rest	1.08
5	High-speed run to the line, sudden turn and return with no rest	0.93–1.11

**Table 2 sensors-23-03179-t002:** Summary of the main experimental set-up, research questions, and the main outcomes.

Experimental Design	Research Question	Main Result
Recorded position	Accuracy of recorded (stationary) position	Better than 0.2 m
Run with transponders around the ankles	Are sudden fluctuations in foot speed recorded?	No. Fluctuations are largely averaged over a full stride cycle
Sudden changes in position	Are sudden changes in position accurately recorded?	No. Errors up to ≈1 m found
Influence of human body	Does a human body influence the recorded position?	No. No detectable effect within the limits of measurements
Influence of nearby transponders	Does nearby transponders influence the recorded position?	No. No detectable effect within the limits of measurements
Calculated results	Are results calculated from measured values reliable?	Yes. No essential error found

## Data Availability

Data (Excel-files) can be provided by the authors upon request.

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
