# Peer review of "Examination of the ZXY Arena Tracking System for Association Football Pitches"

_sensors, 2023, doi:10.3390/s23063179_

Round 1
Reviewer 1 Report
The authors provides examination of a wearable device for positioning in football fields. The reviewer finds it need to address the following issues:
- though with practical needs (i.e., reliable football game statistics), the authors fail to provide a clear description of the problem (e.g., what makes the work scientifically challenging, esp. comparing to related work on the same topic).
- in the methodology section, text style (e.g., listing) should be used to improve readability
- the authors need to highlight the scientific contribution rather than proving solely the results of the examination.
Author Response
We thank the reviewers for their constructive criticism. We have largely modified the manuscript according to their suggestions. At some points their comments are very short and not specific, and we may therefore have misunderstood their arguments.
Answer to reviewer #1
Description of the problem. Throughout the manuscript we now emphasize that the quality of calculated results depends on the quality of the raw data used.
Text style listing in the methodology section. The method chapter has three subsection. In the first on we give and overview of the ZXY system. In the second we have a pointwise list of the experiments carried out. In the third we give a short summary of how the data have been analyzed. Have we misunderstood your comment?
High-light the scientific contribution. We have now rewritten parts of the introduction and discussion to high-light the scientific contribution.
Reviewer 2 Report
The introduction should be broken down into Introduction and Background sections instead. There is one paragraph i.e. the second paragraph which reviewed slightly the approaches to analyzing the games and mentioned that they may involve analyses of the movement of the players. However, the background discussion should be elaborated further to give more depth on the actual limitations that need remedies.
A clear research problem statement/section should be in place before starting with the Methods section. This research problem statement could lead readers to under the limitation(s) that ZXY is designed to solve and how ZXY can supersede the existing techniques used in football games.
The paper in its existing state presented the Methods, without a conceptual framework/model. The framework/model is important to illustrate the novelty of this ZXY system and to prove its worthiness as a proposed solution to mitigate what is missing in the existing footballer tracking system.
It is not clear why derived quantities are needed in this statement "The ZXY-system provides a number of derived quantities from the measured ones. 366 We calculated several of the derived quantities from the measured ones using standard, 367 well-known equations and principles that can be found in textbooks of numerical mathematics (e.g., (24))."
There should be statistical/quantitative comparisons to prove numerically before a statement like this can be made " Nevertheless, the performance of the ZXY system in the present study seems to be among the best as judged from the examinations of instruments discussed above. "
Author Response
We thank the reviewers for their constructive criticism. We have largely modified the manuscript according to their suggestions. At some points their comments are very short and not specific, and we may therefore have misunderstood their arguments.
Answer to reviewer #2
Introduction. We have now reorganized the introduction to include two subsection (background and research issues). We further emphasize our approach (lines 65–73), partly along principle proposed by Rico-Gonzalez et al.
Research problem. We now end the introduction by stating the main research issues.
Methods, conceptual framework. We have now rewritten parts of the methods to highlight this.
Derived quantities. We now address this issue in the introduction.
Quantitative comparisons. We have now added more values (e.g. lines 448–451).
Reviewer 3 Report
Congratulations, it is q great work; I suggest improve the abstract section with more description with respect the final results.
- What is the main question addressed by the research?, please answer this question in abstract section
- References can be updated to newer versions
Author Response
We thank the reviewers for their constructive criticism. We have largely modified the manuscript according to their suggestions. At some points their comments are very short and not specific, and we may therefore have misunderstood their arguments.
Answer to reviewer #3
Abstract. We have now added some more data in the abstract.
Literature review. We have now updated the literature review and address a number of recent relevant studies.
Round 2
Reviewer 1 Report
The submission has managed to address the comment in the 1st round. The reviewer would suggest some additional improvement to further improve the representation of the work.
- in section 2.2, better to add a figure to illustrate the setting of the experiment, i.e., the appearance of the device, the installment of the device, experiment tasks.
- in section 3, better to add a table to summarize the novel experiment settings and the corresponding results.
Author Response
Thank you for your comments.
We have in the Methods now added two figures to explain the experimental set-up, see lines 117-122 and 143-153 and figures 1 and 2
We have now summarized the results in a separate table (tab. 2), see lines 371-375.
